# Inhalable Microparticles Embedding Biocompatible Magnetic Iron-Doped Hydroxyapatite Nanoparticles

**DOI:** 10.3390/jfb14040189

**Published:** 2023-03-28

**Authors:** Eride Quarta, Michele Chiappi, Alessio Adamiano, Anna Tampieri, Weijie Wang, Teresa D. Tetley, Francesca Buttini, Fabio Sonvico, Daniele Catalucci, Paolo Colombo, Michele Iafisco, Lorenzo Degli Esposti

**Affiliations:** 1Department of Food and Drug, University of Parma, Parco Area delle Scienze 27/A, 43124 Parma, Italy; 2National Heart and Lung Institute, Faculty of Medicine, Imperial College London, London SW7 0AZ, UK; 3Institute of Science, Technology and Sustainability for Ceramic Materials (ISSMC), National Research Council (CNR), Via Granarolo 64, 48018 Faenza, Italy; 4Interdepartmental Center for Innovation in Health Products, Biopharmanet_TEC, University of Parma, Parco Area delle Scienze 27/A, 43124 Parma, Italy; 5Institute of Genetic and Biomedical Research (IRGB), National Research Council (CNR), UOS Milan and IRCCS Humanitas Research Hospital, 20089 Rozzano, Italy; 6PlumeStars srl, Parco Area Delle Scienze, 27/A, 43125 Parma, Italy

**Keywords:** hydroxyapatite, superparamagnetic nanoparticles, nanomedicine, microparticles embedding nanoparticles, Trojan microparticles, pulmonary disease, inhalable dry powder

## Abstract

Recently, there has been increasing interest in developing biocompatible inhalable nanoparticle formulations, as they have enormous potential for treating and diagnosing lung disease. In this respect, here, we have studied superparamagnetic iron-doped calcium phosphate (in the form of hydroxyapatite) nanoparticles (FeCaP NPs) which were previously proved to be excellent materials for magnetic resonance imaging, drug delivery and hyperthermia-related applications. We have established that FeCaP NPs are not cytotoxic towards human lung alveolar epithelial type 1 (AT1) cells even at high doses, thus proving their safety for inhalation administration. Then, D-mannitol spray-dried microparticles embedding FeCaP NPs have been formulated, obtaining respirable dry powders. These microparticles were designed to achieve the best aerodynamic particle size distribution which is a critical condition for successful inhalation and deposition. The nanoparticle-in-microparticle approach resulted in the protection of FeCaP NPs, allowing their release upon microparticle dissolution, with dimensions and surface charge close to the original values. This work demonstrates the use of spray drying to provide an inhalable dry powder platform for the lung delivery of safe FeCaP NPs for magnetically driven applications.

## 1. Introduction

Nanoparticles (NPs) have found large applications in medicine and have been used for localized drug regional release, targeted delivery, tumor eradication, imaging and other applications thanks to their unique properties [1]. Recently, NPs have been tested by inhalation for lung disease treatment and diagnosis [2]. Having mucoadhesive or mucus-penetrating properties, NPs can effectively reach the lung tissue, minimizing side effects in comparison to systemic treatments [3].

Usually, NPs for pulmonary diseases are administered through liquid aerosol inhalation. However, nebulizers have intrinsic disadvantages such as limited stability, electric supply and time-consuming delivery; furthermore, NPs may irreversibly aggregate in the liquid dispersion [4]. An inhalable dry powder (dp) formulation bypasses the limits of liquid aerosols, having a longer shelf life, suitable aerodynamic particle size and dose control. A propellant-free dp inhaler device (DPI), activated by the inspiratory air flow rate of the patient, is required for powder aerosolization and inhalation. With this method, specific regions of the lung could be targeted, thus increasing the effectiveness of the treatment, therefore reducing side effects [5].

An innovative inhalation formulation is the development of NPs embedded in microparticles of water-soluble excipients, also referred to as *Trojan microparticles* [4] or nano-in-microparticle systems [6]. With this approach, NPs are protected within the microparticle matrix and are released as microparticles dissolve in the lung lining liquid of lung epithelia. The use of microparticle-embedded NPs together with the advantages of using the dry formulation allows the efficient delivery of NPs to the deep lung.

Superparamagnetic iron oxide (SPION) NPs have been widely employed for many biomedical applications thanks to their distinguished magnetic properties [7]. Indeed, with the use of external magnetic fields, it is possible to guide SPIONs in an organism in order to target a specific site [8]. Inhalable SPIONs were also studied with the aim to guide and hold the NPs in a targeted lung region upon the application of external magnetic fields, increasing the site-specificity of the treatment. SPIONs can be used as a drug delivery system as well as thermo-seeds for magnetic hyperthermia and as contrast agents for magnetic resonance imaging (MRI) [7,9,10]. Previous work has proven the potential of Trojan microparticles embedding SPIONs to magnetically guide NPs toward a specific lung target and undergo hyperthermia-stimulated drug release [11,12,13,14]. However, SPIONs were also proved to be cytotoxic toward epithelial lung cells, posing the risk of triggering inflammatory responses and lung tissue damage [15]. In addition, in vitro and in vivo studies have shown that the intratracheal administration of SPIONs leads to toxicity to the liver and kidneys at both low and high doses, as the NPs can cross the lung–air barrier, spreading systemically in off-site organs [16,17,18]. These data indicate that there is a need for non-toxic inhalable magnetic NP alternatives to SPIONs.

In recent years, our group produced iron-doped calcium phosphate (in the form of hydroxyapatite) nanoparticles (FeCaP NPs) with high magnetic susceptibility and significantly lower cytotoxicity than SPIONs [19,20]. FeCaPs were successfully applied for MRI, drug delivery, hyperthermia-related applications and magnetic field-assisted particle guidance both in vitro and in vivo and thus are an interesting theranostic nanoplatform for lung disease treatment [21,22,23].

Therefore, in this preliminary study, the internalization of FeCaP NPs and their putative cytotoxicity towards human alveolar epithelial cells was studied in vitro. Then, inhalable mannitol microparticles embedding FeCaP NPs (referred to as dp-FeCaPs) were prepared for enhancing FeCaP NPs for lung delivery. A spray drying technique in open mode was used to construct respirable dp-FeCaPs. The D-mannitol was chosen to form the matrix because it is soluble in lung fluid and because it is approved for pharmaceutical use and prescribed via inhalation in non-cystic fibrosis bronchiectasis treatment [24]. Three different ratios between mannitol and FeCaP NPs were designed and studied for manufacturing the dp-FeCaPs. The physico-chemical and technological properties of microparticles were measured, with particular attention to the aerodynamic assessment of dry particle aerosol. A schematic representation of our approach is reported in Figure 1.

## 2. Materials and Methods

### 2.1. FeCaP NPs Preparation and Characterization

All reagents were purchased from Sigma-Aldrich (St. Louis, MO, USA). The synthesis of FeCaP NPs was performed as reported by Iannotti et al. [20]. In detail, 300 mL of a phosphoric acid aqueous solution (H_3_PO_4_, 85 wt.% pure, 20.75 g) was added dropwise into 400 mL of an aqueous suspension of calcium hydroxide (Ca(OH)_2_, 95 wt.% pure, 23.40 g). During the neutralization reaction, 75 mL of an aqueous solution of Fe(II) (FeCl_2_·4H_2_O, 99 wt.% pure, 6.03 g) and 75 mL of an aqueous solution of Fe(III) (FeCl_3_·6H_2_O, 97 wt.% pure, 8.28 g) were added simultaneously. For this reaction, the Fe^2+^/Fe^3+^ molar ratio was set to 1, and the Fe/Ca molar ratio was set at 20 mol%. The whole neutralization reaction was carried out at 45 °C. Once the dropwise addition of all reactants was completed, the solution was kept at 45 °C under constant stirring for 3 h and left to age at room temperature overnight. FeCaP NPs were recovered by centrifugation, extensively rinsed with water, and added to a 0.1 wt.% citrate buffer at pH = 6 to obtain a solution at 10 mg/mL. This suspension was sonicated with a tip sonicator to functionalize the NPs’ surface with citrate. FeCaP NPs were finally recovered by centrifugation, re-suspended in water and stored at 4 °C.

The morphology of FeCaP NPs was investigated using scanning electron microscopy (SEM) by using a Sigma field emission gun SEM microscope (Carl Zeiss, Oberkochen, Germany). A drop of FeCaP NP suspension concentrated at 0.1 mg/mL was deposited on a silicon wafer pre-mounted on aluminum stubs and left to evaporate. Afterward, the sample was sputter-coated with gold in a Sputter Coater E5100 (Polaron Equipment, Watford, Hertfordshire, UK) under argon at 10^−3^ mbar for 1 min with a sputtering current of 30 mA. The micrographs were acquired in secondary electron mode by using a 4 kV acceleration voltage.

The hydrodynamic size and ζ-potential of FeCaP NPs were measured by dynamic light scattering (DLS) with a Zetasizer Nano ZSP instrument (Malvern Ltd., Worcestershire, UK). A disposable electrophoretic cell (DTS1061, Malvern Ltd., Worcestershire, UK) was used for ζ-potential measurements. Measurements were carried out suspending the FeCaP NPs in a 0.01 M 2-[4-(2-hydroxyethyl)piperazin-1-yl]ethanesulfonic acid (HEPES) buffer at pH = 7.4 at the concentration of 0.5 mg/mL. Ten runs of 30 s were performed for each measurement, and four measurements were carried out for each sample.

### 2.2. Analysis of FeCaP NPs Biocompatibility with Human Alveolar Lung Cells

The human transformed type 1 (TT1) cell line was generated as described previously [25] from primary human alveolar type 2 (AT2) cells, which are progenitors of alveolar type 1 (AT1) cells in situ. With this approach, cells are produced that exhibit characteristics of human AT1 cells. Confluent TT1 cell monolayers were incubated with FeCaP NPs suspensions at increasing concentrations (5, 10, 25, 50, 100, 250, 500 and 1000 µg/mL; at 200 mL/well, this equates to 1, 2, 5, 10, 50, 100 and 200 µg/well and approximately 3, 6, 15, 30, 150, 300, and 600 µg/cm^2^ of alveolar epithelium) for 24 h at 37 °C and 5% CO_2_. FeCaP NPs, received as a freeze-dried powder, were suspended in serum-free DCCM-1 culture medium and vortexed prior to being applied to cells. Unexposed cells were used as 100% untreated control (Ctr-) cell viability. Toxic zinc oxide (ZnO) NPs at a final concentration of 15 µg/mL were used as a positive control.

#### 2.2.1. Semi-Quantitative Evaluation of Internalization of FeCaP NPs

The cells were fixed with methanol for 10 min. The cytoplasm was stained with neutral red (pink) and then stained with Prussian blue (blue) to highlight the iron-laden particles. Light microscopy images were generated at 1, 3, 6 and 24 h post-exposure to 5, 10 and 25 µg/mL of FeCaP NPs. Confocal fluorescence microscopy was also used to establish intracellular FeCaP NP localization using 3D image analysis of TT1 cells. TT1 cells were grown to high confluence and exposed to 25 μg/mL FeCaP NPs in serum-free RPMI-1640 medium. Cells were washed to remove apical FeCaP excess then fixed with 4% paraformaldehyde. Samples were then incubated for 15 min at room temperature with wheat germ agglutinin (WGA) stain conjugated to Alexa Fluor 647 (1:100; Invitrogen™, Waltham, MA, USA) to reveal the cell membrane. Cells were washed in PBS and then counter-stained with DAPI (1:100; Invitrogen™, Waltham, MA, USA) for 15 min at RT and rinsed 3 times with PBS before imaging. FeCaP particles were revealed using a 488 nm laser and PMT2 483–496 nm for reflection. Z-Stacks of 3 different samples were taken using the Leica SP5 inverted Confocal Microscope (Leica Microsystems, Wetzlar, Germany) with an optical zoom of 63×. The Z-Stacks depth was chosen in order to include the entire height of the TT1 cells monolayer. Image analysis was performed using the FIJI image analysis software.

#### 2.2.2. Cell Metabolism/Viability Assay

Following treatment, the cell viability was determined as described previously [26]. Washed cells were incubated with 3-(4,5-dimethylthiazol-2-yl)-2,5-diphenyltetrazolium bromide (MTT) solution (500 µg/mL in tissue culture medium) for 25 min at 37 °C and 5% CO_2_. Mitochondrial reductases reduce the MTT to produce purple formazan, which is a measure of metabolic activity and hence measures cell death (reduced activity); an increase in MTT activity indicates the cells have been activated. The formazan blue is solubilized following washing the cells with PBS and the solubilization of the cells with dimethyl sulfoxide (DMSO). The optical density was measured at 570 nm using an Infinite^®^ F50 absorbance microplate reader (Tecan, Nänikon, Switzerland).

#### 2.2.3. Lactase Dehydrogenase Assay (LDH); Reduced Cell Membrane Integrity and Necrotic Cell Death

Forty-five minutes prior to the denoted treatment time, 10% cell lysis buffer was added to replicate untreated cell wells, causing LDH release, for use as a positive control of the total available cellular LDH (referred to as LB Ctr). LDH is released extracellularly as a result of impaired plasma membrane integrity and necrotic cell death. Following cell treatment, the conditioned media were collected, and extracellular leaked LDH levels were determined according to the manufacturer’s instructions (Pierce™ LDH Cytotoxicity Assay Kit, Thermo Fisher Scientific, Waltham, MA, USA). The optical density was measured at 450 nm in an Infinite^®^ F50 absorbance microplate reader (Tecan, Nänikon, Switzerland).

#### 2.2.4. Determination of Intracellular Reactive Oxygen Species (ROS)

ROS was measured at regular intervals over 24 h (1, 3, 6 and 24 h) using a commercial dichlorofluorescin diacetate (H_2_DCFDA) assay. Phorbol myristate acetate (PMA) at 1 µg/mL concentration was used as an ROS-inducing positive control. Intact, adherent cells were incubated with 2′,7′–dichlorofluorescin diacetate (H_2_DCFDA; 25 μM in DCCM-1) at 37 °C in a humidified 5% CO_2_ atmosphere for 30 min. ROS oxidation of H_2_DCFDA results in the formation of fluorescent 2,7-dichlorofluorescin (DCF). Cells were subsequently washed with phosphate-buffered saline (PBS) and solubilized in DMSO. The fluorescence intensity was determined on a FLUOstar OPTIMA microplate reader (BMG Labtech, Ortenberg, Germany), at the *E*_x_/*E*_m_ of 485 nm/530 nm and quantified as relative fluorescence units (RFU).

#### 2.2.5. Measurement of Interleukin 8 (IL-8) Mediator Release

Interleukin 8 (IL-8) release was measured using enzyme-linked immunosorbent assay (ELISA). Lipopolysaccharide (LPS) endotoxin at 0.5 ng/mL concentration was used as a pro-inflammatory positive control. LPS was added to TT1 cells at 0.5 ng/mL suspended in 5% newborn calf serum in DCCM-1 medium. Following treatment, the conditioned media were aspirated from the wells. Measurement of IL-8 release was then determined according to the manufacturer’s instructions (Human CXCL-8 Standard TMB ELISA Development Kit, Peprotech Inc., Rocky Hill, NJ, USA). The optical density was measured at 492 nm in an Infinite^®^ F50 absorbance microplate reader (Tecan, Nänikon, Switzerland).

#### 2.2.6. Statistical Analysis

Data are presented as the mean ± standard deviation (SD). A one-way analysis of variance (ANOVA) followed by Dunnett’s post hoc test was used to assess significant differences between exposure regimens compared to non-treated controls using GraphPad Prism^®^ software. *p*-values < 0.05 were considered statistically significant.

### 2.3. dp-FeCaPs Preparation

dp-FeCaPs were produced using a Büchi Mini Spray Dryer (SD) B-290 (Büchi Laboratory Equipment, Flawil, Switzerland). In detail, 10.0 mg/mL FeCaP NP aqueous dispersions were diluted at 1:10 with ultrapure water. Subsequently, D-mannitol (Parteck^®^ M 200 Mannitol, 98 wt.% pure, Sigma Aldrich, St. Louis, MO, USA) was dissolved in a FeCaP NPs dispersion at the concentration of 0.2, 2 or 4 mg/mL in order to obtain 1:0.2, 1:2 or 1:4 FeCaP NPs/mannitol weight ratios, respectively. The final solid concentration of feed solutions was 1.2 mg/mL, 3 mg/mL and 5 mg/mL, respectively. Mannitol-free dp-FeCaP was also prepared as a control. During the drying process, the dispersions were kept under constant magnetic stirring. The inlet temperature was set at 150 °C, the feed rate at 3.5 mL/min, the atomizing air flow at 601 L/h and the aspiration rate at 35 m^3^/h. A Ø 0.7 mm nozzle was used for the generation of the spray. The resulting outlet temperature was 80 °C. All dp-FeCaP samples were stored at 4 °C in sealed glass vials.

### 2.4. dp-FeCaPs Characterizations

The morphology of dp-FeCaPs was investigated using an SEM Auriga Compact instrument (Zeiss, Jena, Germany). Samples were deposited on the carbon tape placed on the aluminum stubs. The morphology and surface of the microparticles were investigated in plan-view by using a 1 kV electron beam acceleration voltage.

Thermogravimetric analysis (TGA) of the samples was carried out with an STA 449F3 Jupiter device (Netzsch GmbH, Selb, Germany). Thermograms were collected from room temperature to 1100 °C with a ramp of 10 °C/min under air flow using an alumina pan.

X-ray diffraction (XRD) patterns were recorded with a D8 Advance Diffractometer (Bruker, Karlsruhe, Germany) equipped with a Lynx-eye position sensitive detector, using the CuK*α* radiation (*λ* = 1.54178 Å) generated at 40 kV and 40 mA. XRD patterns were acquired in the 10–80° (2*θ*) range with a step size of 0.02° and a scanning speed of 0.5 s. Crystal phase quantification was performed through Rietveld refinement with the software TOPAS5 [27] considering a multiphase system using tabulated atomic coordinates of HA [28] and β and δ polymorphs of crystalline mannitol [29].

Fourier transform infrared (FT-IR) spectra of the samples were collected on a Nicolet 5700 spectrometer equipped with an ATR iD7 accessory (Thermo Fisher Scientific Inc., Waltham, MA, USA). Spectra were collected in ATR mode by the accumulation of 32 scans in the range between 4000 and 400 cm^−1^ with a resolution of 2 cm^−1^.

The particle size of the dp-FeCaP spray-dried samples was determined by the laser diffraction (Malvern Mastersizer 3000, Malvern Instruments, Malvern, UK) method. Briefly, 10 mg of powder was dispersed in 10 mL of 0.1% *w*/*v* Span 85 in cyclohexane (batch number: 1302944, Merck KGaA, Darmstadt, Germany). The dispersion was placed in an ultrasonic bath (Ultrasound bath, USC 300T VWR International, Fontenay-sous-Bois, FR) for 1 min and subsequently analyzed. The particle size distribution was measured with 5% threshold obscuration. Data were expressed as volume diameter of the 10th (Dv_10_), 50th (Dv_50_) and 90th (Dv_90_) percentiles of the particle population.

The in vitro aerodynamic assessment of dp-FeCaPs was studied using the Fast Screening Impactor (FSI) apparatus (Copley Scientific, Colwick, UK). The study of the powder’s aerodynamic performance was carried out in triplicate. All blends were aerosolized using the device RS01^®^ (Plastiape, Lecco, Italy): one HPMC capsule (40 ± 5 mg) size 3 (QUALI-V-I, Qualicaps, SP, Whitsett, NC, USA) was aerosolized using a flow rate of 60 L/min with an aspiration time of 4 s. The FSI separates the particles emitted by the RS01^®^. The ratio between the emitted dose (ED) and metered dose defines the emitted fraction (EF). Finally, the weight of the powder deposited on the FFC filter is the fine particle dose (FPD) of the powder (aerodynamic diameter < 5 mm), while the ratio of the FPD with the ED gives the fine particle fraction (FPF). ED, EF, FPD and FPF are indicators of the powder’s aerodynamic performance.

### 2.5. Restoration of FeCaPs NPs from dpFeCaPs

The analysis of the size and surface charge of NPs released after the dissolution of the microparticles was performed in ultrapure water at pH = 7.4. Different quantities of dp-FeCaPs were dispersed in water in order to have a final FeCaP NPs concentration of 0.5 mg/mL and left to stir at room temperature for 15 min to achieve complete microparticle dissolution. Afterward, DLS and ζ-potential measurements were performed as reported above. A drop of FeCaP NP suspension concentrated at 0.5 mg/mL after dp-FeCaP dissolution was deposited on a GaAs n+ wafer pre-mounted on aluminum stubs and left in the dryer to evaporate overnight before SEM analysis, which was performed as described above.

## 3. Results and Discussion

### 3.1. Characterizations of FeCaP NPs

Superparamagnetic FeCaP NPs characterization was extensively performed in previous works [19,22]. In brief, FeCaP NPs are composed of small, isometric iron-doped hydroxyapatite (Ca_10−x_Fe_x_(PO_4_)_6_(OH)_2_) nanocrystals of about 5–10 nm in size aggregated into needle-like nanoparticles, with length of 70–100 nm and a width of 15–20 nm (Figure 2A). FeCaP NPs present on their surface round-shaped maghemite (γ-Fe_2_O_3_) nanoparticles of 5–15 nm in size formed as a secondary phase during FeCaP NP synthesis (Figure 2A). The amount of iron-doped hydroxyapatite estimated by Rietveld refinement was 94.5 ± 0.7 wt.% and maghemite was 5.5 ± 0.7 wt.%. The quantity of iron inside the apatite structure was determined to be 5.74 ± 0.06 wt.% after subtracting iron content due to the maghemite phase [20,22]. FeCaP NPs have a high magnetic susceptibility and display superparamagnetic behavior at room temperature, with a mass magnetization at saturation close to 8.0 emu/g and coercivity close to zero at temperatures > 100 °K. FeCaP NPs dispersed in HEPES buffer at pH 7.4 have a negative ζ-potential of −21 ± 1 mV and an average hydrodynamic diameter of 179 ± 3 nm with a polydispersity index (PdI) of 0.20 ± 0.02 (Figure 2B), indicating good colloidal stability and suitability for use in biological systems [22].

### 3.2. Biocompatibility of FeCaP NPs towards Lung Tissue Cells

FeCaP NPs have previously been shown to have high biocompatibility toward several cell lines in vitro [22,23], but they have never been tested against lung tissue cells. Consequently, the biocompatibility of the FeCaP NPs on respiratory cells was assessed. In this regard, the main target cells are alveolar epithelial type 1 cells (AT1 cells), which are attenuated and very thin (<0.5 µm deep) to enhance gas exchange at the gas–blood barrier, where they mediate molecular cellular events to maintain alveolar homeostasis, particularly between the gas and blood compartment. AT1 cells account for over 95% of the alveolar epithelial surface area and are the major cell type that inhaled particles will interact with. These cells readily internalize engineered nanomaterials and, for example, can translocate carbon nanotubes from the apical to the basal compartment of the epithelium via active and passive processes [30,31]. Therefore, in order to confirm FeCaP NPs’ suitability for lung delivery, we have evaluated their toxicity and bioreactivity towards an immortalized human AT1 cell-like, human transformed type 1 (TT1) cell line. Intracellular particle localization, cell viability, metabolic activity, intracellular oxidative stress and inflammatory cytokine mediator release were assessed 24 h after administration of increasing concentrations of FeCaP NPs.

The uptake of FeCaP NPs by TT1 cells over 24 h was first investigated using light microscopy and the straightforward Prussian blue staining technique (Figure 3). Prussian blue stains ferric iron blue, highlighting the FeCaP NPs interacting with the cells, which in turn were counterstained with neutral red. Whereas no blue deposits were detected in the negative control at any time, the number of blue deposits in the treated cells increased proportionally to the FeCaP NP concentration and over time (Figure 3A). In fact, blue regions were first detected associated with the TT1 cell monolayer, often in the perinuclear region of the cytoplasm, one hour post-exposure to FeCaP NPs; there was no indication of FeCaP NPs overlying the nuclei of cells. This suggests FeCaP NP uptake and internalization, in particular at 10 and 25 µg/mL concentration after 24 h of exposure (marked by black arrowheads in Figure 3B). Darker blue particle aggregates within TT1 cells began to develop at the maximum concentration of 25 µg/mL after 6 and 24 h of exposure. FeCaP NP internalization into TT1 cells at the concentration of 25 µg/mL after 24 h of exposure was also investigated by confocal fluorescence microscopy (Figure 3C) to confirm the actual intracytoplasmic NPs’ location. Confocal analysis showed the FeCaP NPs clusters (using laser reflection) within the cytoplasm of the TT1 cell monolayer (marked by green arrowheads in Figure 3C), also confirmed by the orthogonal view analysis of the Z-stack images. Most of the FeCaP NP clusters were free in the cytoplasm, confirmed to be accumulating close to the nucleus but never inside it. The orthogonal views analysis of the Z-stack images also showed the presence of some NP groups at the basolateral side of the cell cytoplasm, indicating possible translocation. The mechanism of this latter process will be the focus of a future work.

After treatment with FeCaP NPs for 24 h, the viability of the TT1 cells was determined (Figure 4A). Notably, no significant differences in treated cell viability in comparison to untreated cells only were seen at all with FeCaP NP doses ranging from 5 to 1000 µg/mL, in contrast to the highly toxic 15 µg/mL of the ZnO NP positive control. Interestingly, an increase in cell viability proportional to FeCaP NPs concentration was observed, with a significant maximum at 50 μg/mL (15–20% increase; *p* < 0.001), indicating a potential ability of FeCaP NPs to stimulate cell metabolism which does not involve cell death. This returned to normal levels at the higher doses of FeCaP NPs. The process is dynamic and likely to fluctuate during the 24 h exposure for each of the doses under study. The most important finding is that there was no significant cell death even at high concentrations of FeCaP NPs, and this was supported by the complete lack of LDH release (necrotic cell death; see below) unlike for the cell death induced by ZnO NPs.

Lactate dehydrogenase (LDH) release was quantified, as it is an index reflecting plasma membrane damage due to necrotic cell death that may be related to NP uptake (Figure 4B). LDH release quantification confirmed that FeCaP NPs did not interfere with the membrane integrity, and no overt cell death or cytotoxicity occurred following treatment with any concentration of FeCaP NPs, in contrast to ZnO NPs or the cell lysis buffer positive control. Some cells contain significant amounts of FeCaPs, yet their morphology is unchanged (Figure 3); this supports the observation of no overt toxicity via cell membrane damage or LDH release.

Intracellular reactive oxygen species (ROS) were measured regularly for up to 24 h to capture any early peaks in ROS production that might have subsided by 24 h of FeCaP NP exposure (Figure 4C). Similar to the viability assay results, the quantification of ROS concentration revealed no significant ROS formation associated with FeCaP NP administration at any time interval.

Finally, the release of cytokines was quantified to assess the potential induction of an inflammatory response in TT1 cells by FeCaP NP exposure (Figure 4D). The IL-8 (CXCL-8) cytokine release into cell medium following FeCaP NP exposure was scarcely detectable (50 pg/mL), showing that they did not stimulate IL-8 in a physiologically meaningful manner even at greater doses, and hence no pro-inflammatory stimulation occurred, in contrast to the pro-inflammatory LPS used as a positive control.

Overall, these analyses have shown that FeCaP NPs enter TT1 cells, yet exhibit very low bioreactivity with respect to cell cytotoxicity, mediator release and oxidative stress even if a wide concentration range was used. The amounts of NPs that were applied to the cells per unit area (from 3 to 600 µg/cm^2^ of alveolar epithelium) were very high levels at the top end of the exposure range and are well above those that would be used therapeutically. Therefore, these findings indicate that FeCaP NPs are safe to be used for inhalation delivery to the deep lung and are a better alternative to SPIONs for lung therapy and diagnostics.

### 3.3. Preparation and Characterizations of dp-FeCaPs

To transform the biocompatible FeCaP NPs into respirable dry microparticles, dispersions of NPs with mannitol—this latter employed to form microparticles—were spray dried. In a previous paper, this approach enabled the construction of inhalable mannitol microparticles embedding undoped CaP NPs allowing the release of NPs with original size upon mannitol dissolution [6]. Indeed, the high water solubility of mannitol ensured that microparticles quickly dissolve, releasing the embedded CaP NPs [6].

Three suspensions of FeCaP NPs/mannitol with different weight ratios (i.e., 1:0.2, 1:2, and 1:4) were spray-dried. For comparison, a mannitol-free dp-FeCaP was also prepared. The obtained spray-dried powders had a brown color, did not agglomerate in large clusters and were non-electrostatic.

The composition of dp-FeCaPs with the highest and the lowest mannitol content (i.e., dp-FeCaP 1:4 and 1:0.2, respectively) as well as of dp-FeCaP without mannitol was analyzed by TGA, XRD and FT-IR spectroscopy (Figure 5 and Table 1). Upon heating (Figure 5A), all materials showed a small weight loss occurring between 25 and 150–200 °C due to the dehydration of adsorbed water. Interestingly, dp-FeCaP 1:4 had the lowest content of residual water, suggesting that a higher relative content of mannitol induced more efficient drying. dp-FeCaP 1:4 and 1:0.2 presented a second weight loss between 200 and 400 °C which was attributed to mannitol decomposition, as this is not present in dp-FeCaP without mannitol. The actual mannitol content measured by TGA was close to the nominal value for both samples (Table 1). The FeCaP NP content in the microparticles was attributed to the residual weight above 450 °C (Table 1), as these inorganic NPs are non-thermally degradable up to 1000 °C [21]. The actual FeCaP NPs content is in agreement with the nominal NP content and the FeCaP/mannitol weight ratio. Mannitol decomposition of dp-FeCaP 1:0.2 occurred ca. 100 °C lower than in dp-FeCaP 1:4, and in turn, the mannitol decomposition temperature of dp-FeCaP 1:4 was ca. 50 °C lower in comparison to pure mannitol (ca. 360 °C) [32]. This suggests that FeCaP NPs destabilized mannitol and favored its thermal degradation in proportion to the NP content. XRD patterns of dp-FeCaP samples (Figure 5B) showed both diffraction peaks of poorly crystalline HA as well as those of crystalline mannitol. On the other hand, only HA peaks were present for dp-FeCaP without mannitol. Interestingly, the mannitol peaks in dp-FeCaP 1:4 corresponded to the thermodynamically stable β polymeric form, whereas in dp-FeCaP 1:0.2, to the kinetically stable δ form [29]. This suggests that the ratio between FeCaP NPs and mannitol during the drying process influenced the mannitol crystallization, favoring a polymorphic form over the other. The FeCaP NP and mannitol contents were also estimated by Rietveld refinement of XRD data, obtaining a FeCaP NPs/mannitol weight ratio of 86 ± 1/14 ± 1 for dp-FeCaP 1:0.2 and 19 ± 1/81 ± 1 for dp-FeCaP 1:4. Therefore, the FeCaP/mannitol weight ratio estimated by XRD patterns is in close agreement with TGA data and to nominal ratios. FT-IR spectroscopy (Figure 5C) gave further confirmation on the nature of the materials; as for dp-FeCaP dried without mannitol, only the bands associated with HA were present (bending and stretching of phosphate groups at 550–600 and 950–1100 cm^−1^, respectively), whereas for dp-FeCaP 1:4, all peaks were attributed to mannitol, and dp-FeCaP 1:0.2 presented both species. In all cases, the relative intensity of FT-IR bands was in agreement with the FeCaP/mannitol weight ratio.

The morphology of dp-FeCaP microparticles was examined by SEM (Figure 6). Generally speaking, the shape of microparticles generated by spray drying is greatly affected by the composition of the solution to dry [33]. In our work, the spray drying process led to the formation of round-shaped porous particles, with a geometric size dependent on the FeCaP NP/mannitol ratio. All dp-FeCaPs exhibited a rough surface due to the presence of entangled acicular NPs on the surface and an evident high porosity.

Even small amounts of mannitol in the feed solution drove the formation of microparticles in which mannitol seems not to be embedding NPs, as previously obtained with undoped CaP NPs [6]. Instead, the needle-like FeCaP NPs are all on the surface of the microparticles and are nest-like. In these microparticles, the agglomerate structure is supported by mannitol acting as a binder of NPs.

The dp-FeCaP with the highest mannitol content (i.e., dp-FeCaP 1:4) showed well-separated individual microparticles (Figure 6A). In contrast, it can be observed that with lower or without mannitol content, dp-FeCaPs have a less regular morphology (Figure 6B), presenting some merged microparticles (see the SEM micrographs of dp-FeCaP 1:0.2 and dp-FeCaP without mannitol in Figure 6C,D). Finally, all samples exhibit a large porosity in the microparticles that can be beneficial for their respirability and NP release.

The decrease in the microparticle size of the four formulations was related to the feed concentrations, which can influence the evaporation rate of droplets during the spray drying process. Solid concentrations of the dispersion to be dried higher than 0.1% *w*/*v* led to sprayed droplets in which the ratio between the evaporation rate and the diffusion rate of solute molecules and insoluble NPs was expected to be higher than 1 [33]. This ratio is known as the Peclet number; when it is higher than 1, the formation of porous particles is obtained. Therefore, a lower solvent content and high feed concentrations of dp-FeCaPs with higher mannitol contents further increased the Peclet number values, leading to lower measured geometric particle diameters [34].

The microparticle size distribution of the samples is reported in Table 2. The microparticle size is described by the 10th, 50th and 90th percentiles of volume distribution (Dv_10_, Dv_50_ and Dv_90_, respectively). The size distribution shows that all samples were composed of relatively small microparticles, with 90% of the particle populations smaller than 5–7 μm. Volume diameter values decreased by increasing the mannitol content, with dp-FeCaP 1:4 having the lowest values.

Apart from particle size, the parameter that controls particle penetration in the lung is the aerodynamic particle diameter. Aerodynamic particle size depends on particle geometric size, density and shape. Therefore, for efficient delivery, both particle dimensions and composition must be taken into account [35,36]. In this regard, the use of mannitol as an excipient for Trojan microparticles has the advantage of decreasing microparticle density, improving the respirability of the powder and bypassing the issues bound to inhaling high-density inorganic NPs [4]. In general, the ideal aerodynamic particle diameter that allows particles to reach the pulmonary alveoli in the deep lung ranges from 1 to 5 µm [35,36]. Indeed, particles larger than 5 µm are poorly respirable, whereas ultrafine particles (<1 µm) can remain suspended in the alveoli and be exhaled without deposition [37].

The dp-FeCaP aerodynamic behavior was assessed in vitro using a fast screening impactor (Table 2) to determine (i) the emitted fraction (EF), i.e., the powder extracted during inhalation act from the DPI, and (ii) the aerosol’s fine particle fraction (FPF), having an aerodynamic diameter lower than 5 µm, which represents the respirable fraction. The EF values were higher than 75% and were comparable between samples indicating that all dry powders were successfully emitted from the RS01^®^ Medium Resistance Monodose DPI. The FPF for dp-FeCaP without mannitol was 34.5%. The addition of mannitol to the Trojan microparticle promoted the respirability of the magnetic microparticle over 65% proportionally to the content of the mannitol. This value is significantly higher than the FPF published on similar compositions of microparticles containing SPIONs [11]. In Table 2, the increase in the microparticles’ FPF reflects the decrease in geometric particle size or, in the case of particles having similar shapes, a decrease in apparent density due to higher porosity. The highest FPF value was achieved with the FeCaP/mannitol 1:4 ratio. The high amount of low-density mannitol allowed the production of porous microparticles with smaller sizes and densities.

### 3.4. Release of FeCaP NPs by dp-FeCaPs Dissolution

The hydrodynamic diameter and the surface charge of FeCaP NPs released after microparticle dissolution in water in comparison to their original attributes were analyzed by DLS (Table 3) [38]. This measurement is referred to as the *restoration test* and assessed whether spray drying and the use of mannitol as an excipient preserved the FeCaP NPs’ original quality in terms of size and surface charge after microparticle dissolution in fluids.

In the case of dp-FeCaP 1:0.2, the hydrodynamic diameter of the restored FeCaP NPs had a micrometric value, assigned to the lowest mannitol content in the microparticles causing NP aggregation during the drying process. The use of higher amounts of mannitol prevented FeCaP NP aggregation and resulted in hydrodynamic diameters, after restoration, close to the original values. The smallest hydrodynamic diameter, as well as the most negative ζ-potential, were obtained for dp-FeCaP 1:4, indicating that the 80 wt.% of mannitol in the composition preserved the FeCaP NPs properties enabling release in the deep lung NPs at their original quality. The ζ-potential of FeCaP NPs obtained by dissolution of dp-FeCaP 1:4 is more negative in comparison to the original NPs (−31 vs. −21 mV). The increase in surface charge might be due to the presence of a high amount of mannitol in the medium after the microparticles’ dissolution. FeCaP NPs, after the restoration of dp-FeCaP 1:2, were analyzed by SEM (Figure 7), showing that their needle-shaped morphology was preserved after microparticle dissolution.

## 4. Conclusions

We proved that superparamagnetic FeCaP NPs are not cytotoxic towards human lung alveolar epithelial cells, even at relatively high concentrations, which supports the concept that they are safe to be used for inhalation in lung applications.

To protect NPs’ qualities (i.e., size and surface charge) and provide inhalable dry powders, microparticles have been manufactured by spray drying an aqueous dispersion of FeCaP NPs and D-mannitol, which was employed to form a matrix to embed NPs. Differently from what was previously reported with undoped CaP NPs [6], FeCaP NPs were not embedded into microparticles but distributed on the dp-FeCaPs’ surface. Our data prove that the mannitol and FeCaP NP ratio in the feed suspension was crucial for manufacturing microparticles with large porosity and an aerodynamic size of less than 5 µm, achieving products with excellent respirability. Upon the dissolution of microparticles, FeCaP NPs can be released at the original nano-size and surface charge, allowing them to possibly exert their action on the lung tissue or to cross the air–blood barrier. Therefore, with our nanoparticles-in-microparticles approach, we obtained a safe inhalable product for delivering magnetic NPs to the lung for therapy, diagnostic, or theranostic applications.

Overall, our inhalable powders carrying superparamagnetic FeCaP NPs represent the proof of concept for a new nano-engineered material that could substitute the more toxic SPION NPs for the lung delivery of magnetic materials. In future work, we will explore the suitability of inhalable dp-FeCaP to be used for simultaneous MRI, magnetotherapy and drug delivery purposes in the lung to treat pathologies such as cancer or cystic fibrosis.

## Figures and Tables

**Figure 1 jfb-14-00189-f001:**
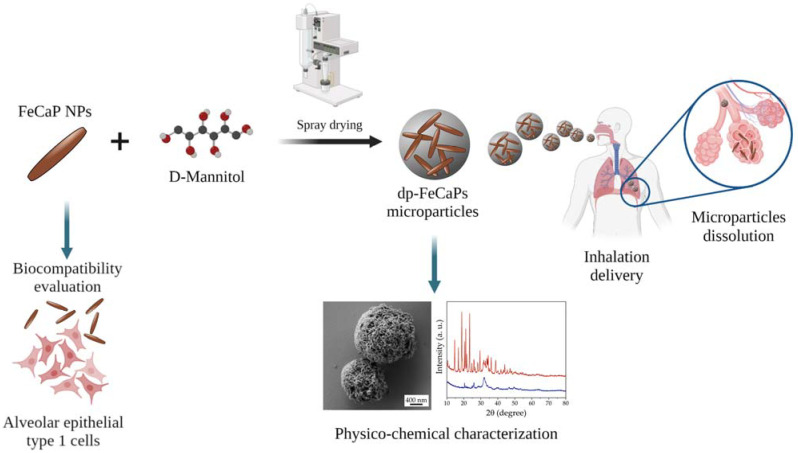
Schematic representation of the fabrication of inhalable dp-FeCaP microparticles embedding FeCaP NPs and their potential application.

**Figure 2 jfb-14-00189-f002:**
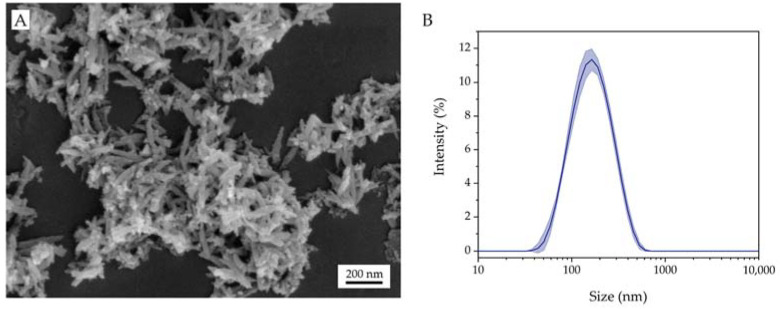
(**A**) SEM micrograph of FeCaP NPs. (**B**) Hydrodynamic size distribution by intensity of FeCaP NPs.

**Figure 3 jfb-14-00189-f003:**
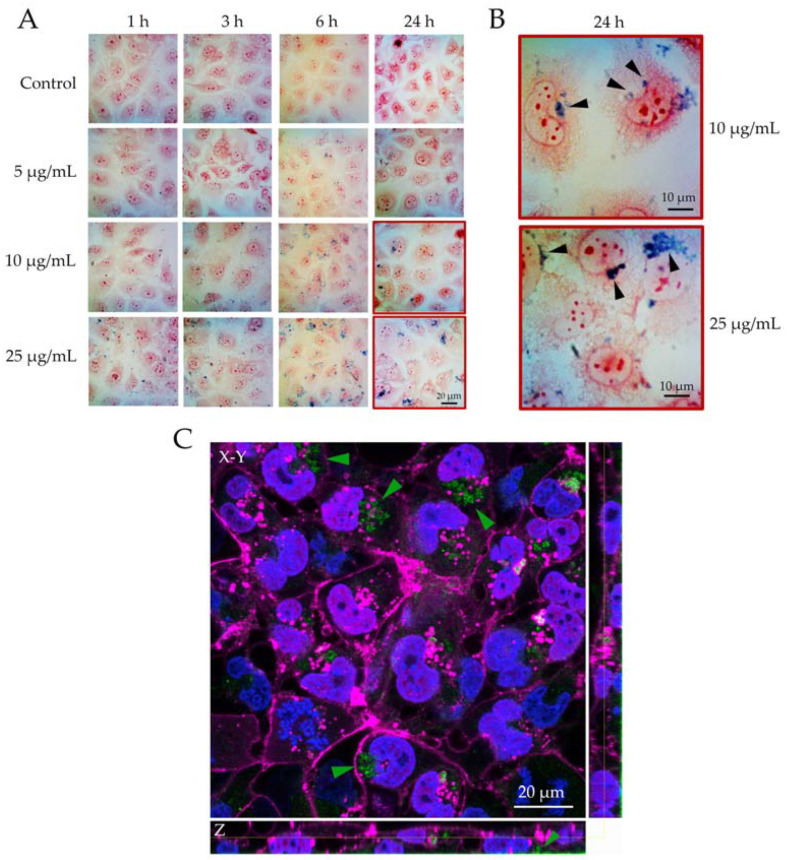
(**A**,**B**) Light microscopy and (**C**) confocal microscopy images of FeCaP NPs internalized into TT1 cells. (**A**) Visualization of TT1 cells at increasing FeCaP NP exposure concentrations for up to 24 h. (**B**) Enlarged section of cells from the two highest exposure concentrations at 24 h, highlighting the cell-associated FeCaP NPs (marked by black arrowheads). Cytoplasm—pink/red; FeCaP NPs—blue. (**C**) Confocal microscopy image of TT1 cells at 24 h exposure to 25 µg/mL FeCaP NPs. Cell membranes—pink; nuclei—blue; FeCaP NPs—green. Green arrowheads highlight FeCaP NP clusters within the cytoplasm. Yellow line represents the Z confocal plane of the X–Y view (i.e., the single slice from the stack that is shown in the square image). Light microscopy images were taken using an optical zoom of (**A**) 20× or (**B**) 40×; confocal microscopy images were taken using a zoom of 63×. Scale bar: 20 µm.

**Figure 4 jfb-14-00189-f004:**
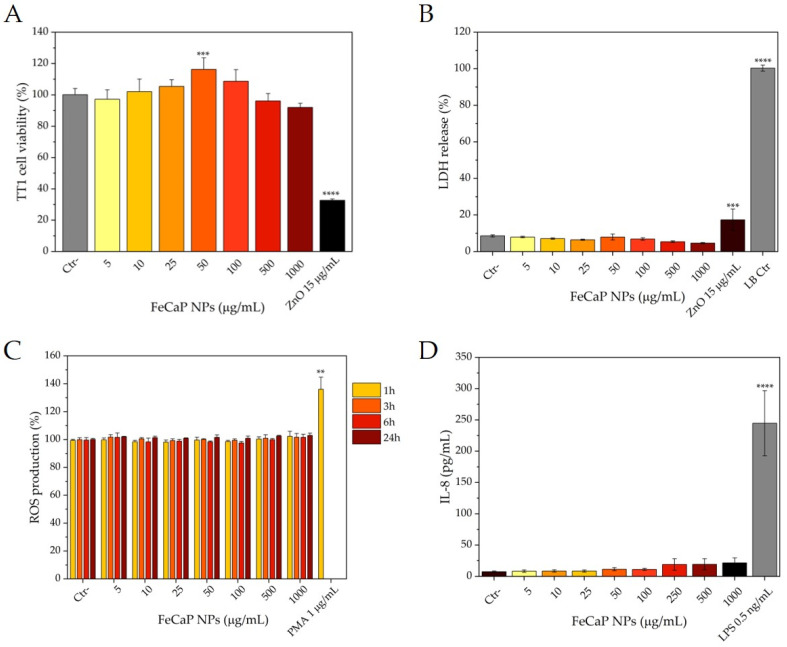
Effect of FeCaP NPs on cell activity and viability (MTT and LDH assays), cellular reactive oxygen species (ROS) production and IL-8 mediator release. TT1 cells were exposed to increasing concentrations of FeCaP NPs and analyzed after 24 h of exposure (MTT, LDH, IL-8) or at 1, 3, 6 and 24 h after exposure (ROS). ZnO NPs at the concentration of 15 µg/mL were used as a negative control for MTT assay, and as a positive control for LDH assay together with lysis buffer. PMA at the concentration of 1 µg/mL was used as positive control for ROS assay, and LPS at the concentration of 0.5 ng/mL was used as a positive control for IL-8 release assay. (**A**) MTT assay measures the metabolic activity of the cells, indicating cell activation (increased) and cell death (decreased). (**B**) LDH measures necrotic cell death with cell membrane damage and extracellular leakage of LDH. (**C**) Intracellular ROS indicates levels of oxidative stress, performed at regular intervals up to 24 h to capture any peaks of activity. (**D**) IL-8 mediator release indicates induction of proinflammatory activity. N = 3 experiments performed in triplicate. **, *p* < 0.005; ***, *p* < 0.001; ****, *p* < 0.0001.

**Figure 5 jfb-14-00189-f005:**
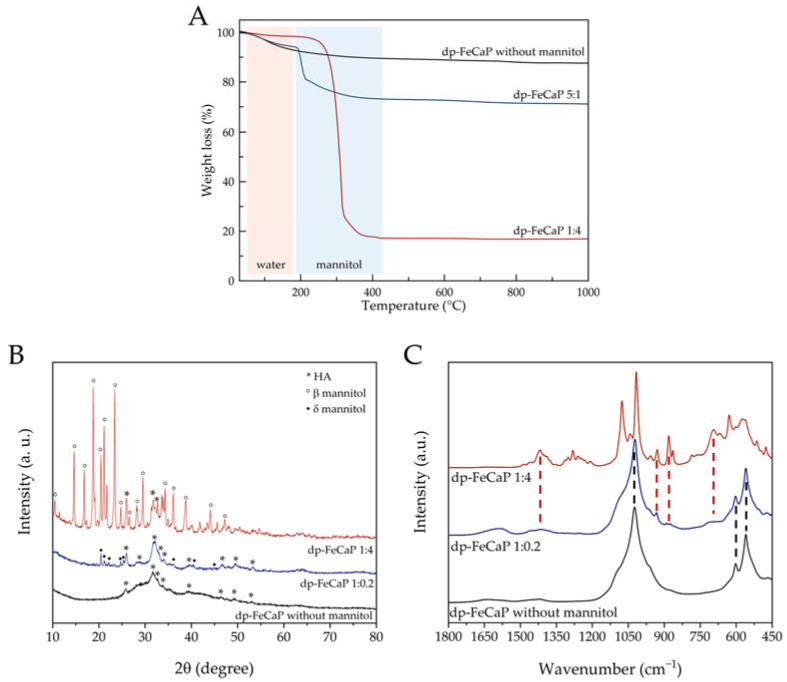
(**A**) TGA curves, (**B**) XRD patterns and (**C**) FT-IR spectra of dp-FeCaPs produced with FeCaP NPs/mannitol ratio of 1:0.2, 1:4 and without mannitol.

**Figure 6 jfb-14-00189-f006:**
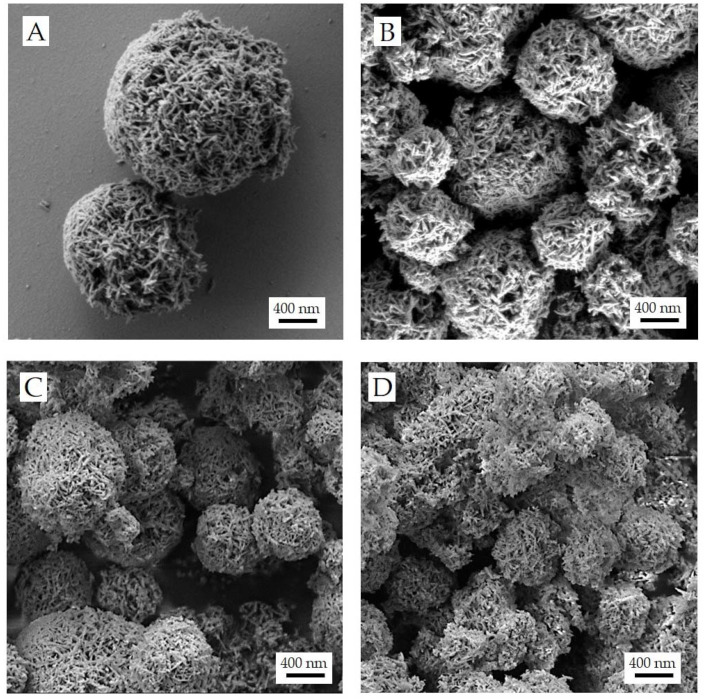
SEM micrographs of dp-FeCaPs produced with a FeCaP NPs/mannitol ratio of 1:4 (**A**), 1:2 (**B**), 1:0.2 (**C**) and without mannitol (**D**).

**Figure 7 jfb-14-00189-f007:**
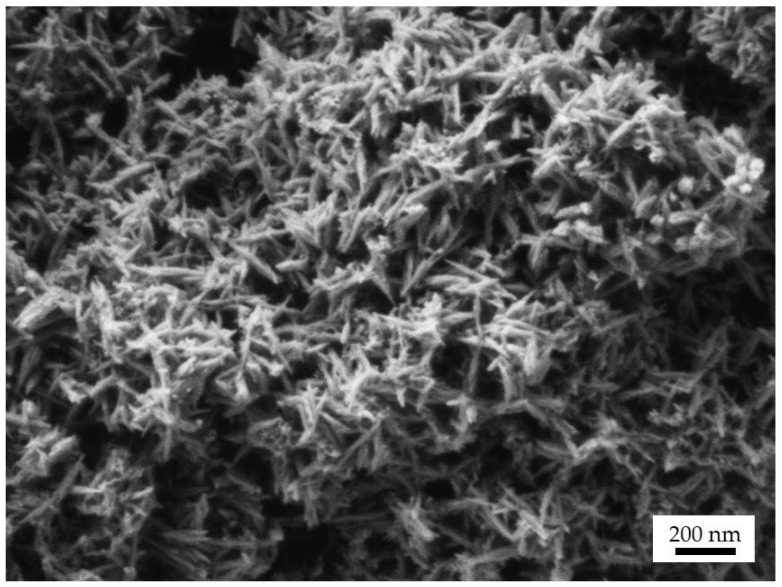
SEM micrograph of released FeCaP NPs after dissolution in water of dp-FeCaP 1:2.

**Table 1 jfb-14-00189-t001:** Composition of dp-FeCaPs quantified by TGA.

Sample	Nominal Mannitol Content (wt.%)	Water (wt.%)	Mannitol (wt.%)	Residual at 450 °C (FeCaP NPs) (wt.%)
dp-FeCaP without mannitol	-	8.7 ± 0.9	-	89.4 ± 9.0
dp-FeCaP 1:0.2	16.7	5.2 ± 0.5	21.7 ± 2.0	73.1 ± 7.0
dp-FeCaP 1:4	80	1.4 ± 0.1	81.2 ± 8.0	17.2 ± 2.0

**Table 2 jfb-14-00189-t002:** Particle size distribution expressed as 10th, 50th and 90th percentiles of volume distribution (Dv_10_, Dv_50_ and Dv_90_) and in vitro aerodynamic parameters including the emitted fraction (EF) and fine particle fraction (FPF) of the dp-FeCaPs. All values are reported as mean ± standard deviation (n = 3).

Sample	Dv10 (μm)	Dv50 (μm)	Dv90 (μm)	EF (%)	FPF (%)
dp-FeCaP without mannitol	1.20 ± 0.02	3.32 ± 0.01	7.22 ± 0.05	75.2 ± 0.3	34.5 ± 01
dp-FeCaP 1:0.2	0.90 ± 0.02	2.70 ± 0.01	6.91 ± 0.04	75.6 ± 0.2	44.6 ± 0.2
dp-FeCaP 1:2	0.65 ± 0.01	2.27 ± 0.01	6.60 ± 0.05	75.4 ± 0.1	60.3 ± 0.1
dp-FeCaP 1:4	0.60 ± 0.01	2.22 ± 0.02	5.68 ± 0.09	76.3 ± 0.1	65.7 ± 0.1

**Table 3 jfb-14-00189-t003:** Particle size and ζ-potential values of released FeCaP NPs from dp-FeCaPs after dissolution in water. Values of original FeCaP NPs are also reported as a comparison.

Sample	Hydrodynamic Diameter (nm)	PdI	ζ-Potential (mV)
Original FeCaP NPs	179 ± 3	0.20 ± 0.02	−21 ± 1
dp-FeCaP 1:0.2	1105 ± 24	0.30 ± 0.01	−22 ± 1
dp-FeCaP 1:2	243 ± 4	0.24 ± 0.02	−23 ± 1
dp-FeCaP 1:4	212 ± 1	0.26 ± 0.02	−31 ± 1

## Data Availability

The data presented in this study are available in the article.

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
