# Peer review of "Inhalable Microparticles Embedding Biocompatible Magnetic Iron-Doped Hydroxyapatite Nanoparticles"

_jfb, 2023, doi:10.3390/jfb14040189_

Round 1

Reviewer 1 Report

In this manuscript, the authors proposed a nanoparticle-in-microparticle approach of calcium phosphate, and demonstrated their properties for pulmonary delivery. This work seems to be interesting and useful. However, the following problems should be addressed before further consideration of publication:

1. The title is suggested to be more clear and brief without ambiguity. A scheme can be created after the Introduction section to better demonstrate the sample and process.

2. All the abbreviated terms should be stated in detail at the first appearance, such as PDI, PdI, HEPES.

3. All the figures need to be revised with higher resolutions since some labels are not clear. The increase at first and then decrease in Figure 3A need to be explained. In Figure 3D, the range of Y coordinates can be reduced to show the details clearly.

4. The material characterization was relatively simple. EDS, TEM/HRTEM, and XRD analysis can be added for the samples, which can help explain the hybrid structure.

5. Table. 2 is really confusing and needs to be explained in detail. What’s the significance to conduct particle size statistics, and how can we ensure the measurement accuracy in this case?

6. The references should be checked including some format errors (e.g., needless DOI, ref. 15). Recent advances in targeted therapy can be added (e.g., 10.1021/acsami.1c16859).

Reviewer 2 Report

Comments:

1.     In Figure 2c, the author stated that the FeCaP NPs clusters were surrounded within endosome, I think this confocal image is too qualitatively to show this data, I would recommend the authors stained the endo/lysosome of the cells and study if the FeCaP NPs inside the endosome.

2.     How about the endosomal escape ability of this type of FeCaP NPs? If they are kept inside the endosome, I would expect there will be very low efficacy of using this FeCaP NP for drug delivery.

3.     I also curious about the cell association of this FeCaP NPs, how much percentage of the FeCaP NPs can be uptake by the TT1 cells, if there is very low percentage of FeCaP NPs uptake by the cells, there will be also negligible cell toxicity observed in Figure 3A.

4.     What is the encapsulation efficiency of the LDH, or IL-8 of the FeCaP NPs? I think the author should provide this.

5.     What is the pore size distribution of the dp-FeCaP microparticles? I would recommend the author provide some BET data to show this.

6.     In Figure 6, for the dp-FeCaP, it seems that the dp-FeCaP cannot be stable in water, I was curious how the author stored those dp-FeCaP, and I think it will be very hard for biological applications since the dp-FeCaP will be immediately dissolved after the inhalation. Can the author provide some comments to that?  

7.     I would also recommend the author provide some in vitro or in vivo model to show that after the inhalation, the dp-FeCaPN NPs can still release the cytokine or other drugs and can have good efficacy.

Reviewer 3 Report

We appreciate your work but we suggest revision as no coherence about flow and need to justify your approach why this is better.

1.What is novelty for this Inhalable dry powder microparticles embedding superparamagnetic calcium phosphate nanoparticles for pulmonary delivery?

2.What kind of  disease targeted with your approach ?

3.How your approach better than conventional available strategy?

4.Do we need theranostic type of delivery for pulmonary delivery ?

5.Why did you perform TGA ? as it showed high % residue remained at high temperature,how it corelated with your applications.

Round 2

Reviewer 1 Report

I'm satisfied with the revised manuscript.

Reviewer 2 Report

Overall, all the comments have been addressed. I would recommend to accept the current version.